# STING-Mediated Autophagy Is Protective against H_2_O_2_-Induced Cell Death

**DOI:** 10.3390/ijms21197059

**Published:** 2020-09-25

**Authors:** Amar Abdullah, Frank Mobilio, Peter J. Crack, Juliet M. Taylor

**Affiliations:** Department of Pharmacology and Therapeutics, University of Melbourne, Parkville, Melbourne 3010, Australia; amar.abdullah@unimelb.edu.au (A.A.); mobiliof@unimelb.edu.au (F.M.); pcrack@unimelb.edu.au (P.J.C.)

**Keywords:** STING, interferon, autophagy, oxidative stress

## Abstract

Stimulator of interferon genes (STING)-mediated type-I interferon signaling is a well characterized instigator of the innate immune response following bacterial or viral infections in the periphery. Emerging evidence has recently linked STING to various neuropathological conditions, however, both protective and deleterious effects of the pathway have been reported. Elevated oxidative stress, such as neuroinflammation, is a feature of a number of neuropathologies, therefore, this study investigated the role of the STING pathway in cell death induced by elevated oxidative stress. Here, we report that the H_2_O_2_-induced activation of the STING pathway is protective against cell death in wildtype (WT) MEFSV40 cells as compared to STING^−/−^ MEF SV40 cells. This protective effect of STING can be attributed, in part, to an increase in autophagy flux with an increased LC3II/I ratio identified in H_2_O_2_-treated WT cells as compared to STING^−/−^ cells. STING^−/−^ cells also exhibited impaired autophagic flux as indicated by p62, LC3-II and LAMP2 accumulation following H_2_O_2_ treatment, suggestive of an impairment at the autophagosome-lysosomal fusion step. This indicates a previously unrecognized role for STING in maintaining efficient autophagy flux and protecting against H_2_O_2_-induced cell death. This finding supports a multifaceted role for the STING pathway in the underlying cellular mechanisms contributing to the pathogenesis of neurological disorders.

## 1. Introduction

Type-I interferons (IFNs) are pleiotropic cytokines that have been implicated in neuropathologies, including Gaucher disease [1], Aicardi–Goutieres syndrome [2] and a model of prion disease [3]. Supporting this, several reports have identified a detrimental role for type-I IFNs in animal models of Alzheimer’s disease (AD) [4,5], Parkinson’s disease (PD) [6] and traumatic brain injury (TBI) [7], with elevated expression of type-I IFNs found in post-mortem human AD, PD and TBI brains [4,6,7]. Classically, type-I IFNs (IFN-α and IFN-β) mediate their proinflammatory effects through the Janus activated kinases (JAK)-signal transducer and activator of transcription (STAT) pathway. Following binding to their cognate receptor, composed of interferon receptor 1 (IFNAR1) and interferon receptor 2 (IFNAR2) subunits, type-I IFNs signal through this pathway to induce the upregulation of type-I IFN and other proinflammatory cytokine gene expression.

Type-I IFNs can alternatively be activated through the stimulator of interferon genes (STING)-dependent pathway. Foreign materials released by invading pathogens, including DNA or self-derived nucleic acids from dying cells, known as damage-associated molecular patterns (DAMPs), have been shown to activate STING. Through a signaling cascade involving tumor necrosis factor (TNF) receptor-associated factor NF-κB activator (TANK)-binding kinase 1 (TBK1) and interferon regulatory factor-3 (IRF3), this trigger increases type-I IFN production. Aberrant STING-mediated type-I IFN production has been implicated in autoinflammatory diseases including vascular and pulmonary syndrome and lupus [8,9]. STING has also been found to be upregulated in radiation-induced liver injury [10], oxidative stress-induced DNA damage [11] and a mouse model of chronic obstructive pulmonary disease (COPD) [12], highlighting that activation of this pathway is beyond the infection setting. More recently, STING activation has been reported through the release of mitochondrial DNA (mtDNA) in acute kidney injury [13], and we have demonstrated a proinflammatory role for this pathway in a mouse model of TBI [7], further supporting a critical role for STING in DAMP-associated disease pathologies. However, the underlying mechanisms that trigger the STING mediated-type-I IFN-mediated inflammatory response under pathophysiological stress, including elevated oxidative stress, are still not well understood.

Several studies have proposed a link between STING-mediated type-I IFN production and elevated cellular reactive oxygen species (ROS) levels in an infection setting. In one study, 5,6-dimethylxanthenone-4-acetic acid (DMXAA) was found to induce type-I IFN production through the STING pathway [14]. With an earlier study reporting that N-acetylcysteine (NAC) decreases DMXAA-induced proinflammatory cytokine production [15], this potentially links the STING and type-I IFN pathways in the regulation of oxidative stress. A study by Gehrke et al. (2013) implicated STING in increased type-I IFN expression, stimulated by oxidized self-DNA released from dying cells with increased ROS levels correlating with type-I IFN levels [16]. Furthermore, type-I IFN signaling has been shown to be detrimental by inducing oxidative stress with attenuation of this pathway protective in a mouse model of type-I diabetes [17], in a chronic hepatitis virus infection [16,18] and in bacterium-infected macrophages [19]. mtDNA damage is known to contribute to increased intracellular ROS-induced oxidative stress [20,21] with evidence for mtDNA acting as a DAMP molecule to induce STING mediated type-I IFN production [22,23,24]. However, the underlying mechanisms involving the STING and type-I IFN pathways in the context of elevated oxidative stress have not yet been elucidated.

Autophagy is a well characterized cellular degradation and/or recycling process that has been implicated in a number of neuropathologies [25]. Evidence in the literature reports increased expression of autophagy markers by elevated oxidative stress, with both protective and detrimental effects observed [26,27,28,29]. This double-edged sword role of autophagy may be due to the lack of understanding of both the mechanisms and the cell-type specificity under these stress conditions. Interestingly, analogous to its role in protecting cells from invading pathogens, STING has also been shown to be involved in autophagosome formation, a critical step in the autophagy process. Indeed, emerging evidence has shown that DNA viruses and intracellular bacteria can induce autophagy through STING pathway activation [30,31,32]. Double-stranded DNA (dsDNA), including poly (dA:dT) and poly (dG:dC), are also known to trigger autophagy activation [33], supporting a role for STING as an autophagy activator through its DNA sensing ability. Recently, a direct interaction between the autophagy protein LC3 and STING has been reported which is critical for its regulation of autophagy [34]. Furthermore, modulation of the STING/TBK1/IRF3 pathway has been shown to occur, in part, through STING degradation via p62-dependent selective autophagy [35]. These studies suggest that the anti-microbial response and autophagy activation via STING is a tightly controlled event to prevent an excessive inflammatory response in cells.

The activation of the STING-mediated type-I IFN pathway and its contribution to autophagy induced by oxidative stress is not yet understood. This study investigated this using a cell-based model of H_2_O_2_-induced cell death, and here, we report, for the first time, a critical role for STING in mediating the type-I IFN production and increased autophagy flux under oxidative stress conditions. STING^−/−^ cells subjected to H_2_O_2_ treatment displayed a reduced cellular viability as compared to WT cells. Importantly, this protective effect can be attributed, in part, to reduced ROS levels and increased autophagy flux mediated by STING in the WT cells following H_2_O_2_ treatment. Furthermore, the sustained and elevated expression of autophagy markers, including LC3, p62 and LAMP2 levels in H_2_O_2_-treated STING^−/−^ cells, suggests inefficient autophagy flux. Collectively, this study has identified a beneficial role for STING in protecting against H_2_O_2_-induced cell death and proposes STING as a potential target for therapeutic intervention in neuropathologies with oxidative-stress mediated cellular injury.

## 2. Results

### 2.1. The STING Pathway Is Activated Following H_2_O_2_ Treatment

H_2_O_2_ has been used experimentally as a potent ROS inducer, rendering cells into a heightened oxidative stress state, such as that which occurs in a number of neuropathologies. However, the underlying mechanisms that contribute to oxidative stress-mediated cell death are complex, and the roles of the STING and type-I IFN pathways have not been investigated. To confirm a possible role for STING in H_2_O_2_-induced oxidative stress, mRNA expression was analyzed by qPCR analysis. STING expression was induced by 500 μM H_2_O_2_ in WT MEF SV40 cells in a time-dependent manner, with increased STING mRNA levels as early as 30 min (2.06 ± 0.34 fold; n.s. *p* = 0.2808) after H_2_O_2_ treatment as compared to the control group. Interestingly, a second wave of increased STING mRNA expression was also detected at 24 h (1.96 ± 0.72; n.s *p* = 0.3958) of H_2_O_2_ treatment (Figure 1A). Furthermore, Western blot analysis confirmed increased phosphorylation of the STING (pSTING) protein at 30 min (3.43 ± 0.68 fold; n.s. *p* = 0.8656) and 6 h (5.058 ± 1.712 fold; * *p* < 0.05) (Figure 1B,C) after H_2_O_2_ treatment as compared to the untreated control group. However, pSTING expression was unchanged in cells treated for 12–48 h 500 μM H_2_O_2_ treatment (data not shown).

Downstream of STING activation, pTBK1 (Figure 2A) and TBK1 (Figure 2B) protein expression was also determined by Western blot analysis, with increased pTBK1 expression identified in WT cells at early time points (15 min–6 h) of H_2_O_2_ treatment but not in the STING^−/−^ cells. A significant increase in pTBK1 levels was detected at 30 min (4.65 ± 1.43 fold; * *p* < 0.05) and 1 h (4.629 ± 1.33 fold; * *p* < 0.05) (Figure 2C) after H_2_O_2_ treatment as compared to their genotype control group, suggesting H_2_O_2_-induced pTBK1 expression is STING-mediated. Collectively, these results confirmed our hypothesis that the STING pathway is activated after H_2_O_2_ treatment.

### 2.2. H_2_O_2_ Induces Type-I IFN Signalling in a STING-Dependent Manner

To further characterize the activation of the STING pathway following H_2_O_2_ insult, mRNA expression of IRF3, IRF7, IFNα and IFNβ were measured by qPCR analysis. Consistent with STING activation, increased IRF3 mRNA levels were identified in WT cells after H_2_O_2_ treatment, with a reduced response in STING^−/−^ cells (Figure 3A). IRF3 was significantly reduced in STING^−/−^ cells at 48 h of H_2_O_2_ treatment as compared to the WT cells (WT = 4.24 ± 1.22 vs. STING^−/−^ = 1.079 ± 0.22; ** *p* < 0.01). In contrast, increased IRF7 mRNA expression was induced by H_2_O_2_ in both WT and STING^−/−^ cells with no significant difference between these two genotypes (Figure 3B). These results suggest H_2_O_2_-induced IRF3 expression is STING-mediated while IRF7 can be activated independent of STING after H_2_O_2_ treatment. A robust upregulation in IFNα and IFNβ levels was identified in WT cells but reduced in the STING^−/−^ cells at early time points (15 min–6 h) of H_2_O_2_ treatment. Peak activation of IFNα was detected at 1 h (WT = 43.32 ± 6.40 vs. STING^−/−^ = 3.953 ± 1.53; *** *p* < 0.001) and 2 h (WT = 90.23 ± 18.17 vs. STING^−/−^ = 5.65 ± 1.79; *** *p* < 0.001) in WT cells, with this upregulation reduced in the STING^−/−^ cells (Figure 3C). Similarly, IFNβ levels were significantly increased in WT cells at 2 h (WT = 24.92 ± 4.85 vs. STING^−/−^ = 14.66 ± 3.13; * *p* < 0.05) compared to STING^−/−^ cells following H_2_O_2_ treatment (Figure 3D).

### 2.3. STING Is Required for Cellular Survival Following H_2_O_2_ Insult

The role of STING in cellular death induced by H_2_O_2_-induced oxidative stress was assessed in a concentration- and time-dependent manner. Firstly, WT and STING^−/−^ cells were exposed for 24 h with H_2_O_2_ in a dose-dependent manner (300 μM, 500 μM and 1 mM) and cell viability was assessed by an MTT assay. STING^−/−^ cells were more susceptible to cell death induced by 300 μM (% cell viability of WT = 71.04 ± 2.02 vs. STING^−/−^ = 34.58 ± 0.67; *** *p* < 0.001) and 500 μM H_2_O_2_ (% cell viability of WT = 49.87 ± 0.95 vs. STING^−/−^ 31.20 ± 1.19; *** *p* < 0.001) as compared to WT cells (Figure 4A). A dose of 1 mM of H_2_O_2_ was seen to be extremely toxic to the cells of both genotypes with almost 100% of cells killed after a 24 h incubation (% of cell viability of WT = 0.15 ± 0.01 vs. STING^−/−^ = 0.05 ± 0.001; ns *p* > 0.9999). This result suggests that STING is protective in vitro following H_2_O_2_ treatment. Accordingly, the next experiment was designed to determine the time-dependent toxicity effects of H_2_O_2_-induced cell death in WT and STING^−/−^ MEF SV40 cells. Supporting the previous results, it was found that STING confers protection after H_2_O_2_-induced cell death with 24 h (% of cell viability of WT = 55.42 ± 0.37 vs. STING^−/−^ = 31.59 ± 0.89; ** *p* < 0.05) and 48 h (% of cell viability of WT = 71.17 ± 8.67 vs. STING^−/−^ = 31.59 ± 0.89; ** *p* < 0.01) of H_2_O_2_ treatment, showing a significant reduction in cellular viability in the STING^−/−^ cells as compared to the WT cells (Figure 4B).

### 2.4. STING Is a Negative Regulator of H_2_O_2_-Induced ROS Production

Generation of intracellular ROS following H_2_O_2_ treatment is known to contribute to oxidative stress [36,37,38]. To confirm that H_2_O_2_ treatment leads to ROS generation and to further understand the modulation of the H_2_O_2_-induced oxidative stress by STING in this study, intracellular ROS levels in H_2_O_2_ treated WT and STING^−/−^ cells were measured by a 2′, 7′-dichlorofluorescin (DCF) assay. Increased ROS levels were confirmed in both WT and STING^−/−^ cells after H_2_O_2_ treatment as compared to the untreated control group. Importantly, STING^-/^cells exhibited higher ROS levels as compared to the WT cells with a rapid burst of intracellular ROS production detected at 15 min (WT = 13.55 ± 1.67; n.s. *p* = 0.9666 and STING^−/−^ = 19.23 ± 0.32; * *p* < 0.05) that gradually decreased over H_2_O_2_ treatment (Figure 5). This suggests a role for STING as an inhibitor of H_2_O_2_-induced oxidative stress. Further, we confirmed increased ROS generation induced by H_2_O_2_ indicative of elevated oxidative stress in this in vitro model of inflammation.

### 2.5. STING Is Required to Maintain a Normal Autophagy Flux in Response to H_2_O_2_ Treatment

We demonstrated that STING is required to promote cellular survival in response to H_2_O_2_ treatment with increased intracellular ROS levels identified in the absence of STING. To investigate whether autophagy plays a role in mediating the protective effects of STING after H_2_O_2_ treatment, we examined LC3, p62 and LAMP2 expression profiles in WT and STING^−/−^ cells treated with 500μM H_2_O_2_. Conversion of LC3-I to LC3-II is representative of increased autophagy activation [39] while the degradation of p62 and LAMP2 at the late step of the autophagy process serves as a marker for normal autophagic flux. Western blot analysis identified increased LC3-II levels at earlier time points in both WT and STING^−/−^ cells as compared to the control levels, indicating that autophagy is activated in response to H_2_O_2_ treatment (Figure 6A). Interestingly, a prolonged H_2_O_2_ treatment (24 h and 48 h) (Figure 6B) induced higher LC3-II expression in the absence of STING (24 h = 2.311 ± 0.210 vs. vehicle; n.s *p* > 0.9999, 48 h = 1.975 ± 0.529 vs. vehicle; n.s *p* > 0.9999) compared to WT cells (24 h = 0.964 ± 0.159 vs. vehicle; n.s *p* > 0.9999, 48 h = 1.058 ± 0.258 vs. vehicle; n.s *p* > 0.9999) (Figure 6C). Next, a trend for increased p62 levels was detected in both WT and STING^−/−^ cells, with knockout cells showing a trend for higher and sustained expression after H_2_O_2_ treatment as compared to the WT cells (Figure 6D,E), although this was not significant. It is also noteworthy that a trend of reduced p62 levels at earlier time points (15 min–2 h) in the WT cells (15 min = 0.857 ± 0.259 vs. vehicle; n.s *p* > 0.9999, 30 min = 0.610 ± 0.135 vs. vehicle; n.s *p* > 0.9999, 1 h = 0.992 ± 0.301 vs. vehicle; n.s *p* > 0.9999, 2 h = 0.963 ± 0.282 vs. vehicle; n.s *p* > 0.9999) was detected as compared to the elevated p62 levels observed in the STING^−/−^ cells at similar time points (15 min = 1.200 ± 0.213 vs. vehicle; n.s *p* > 0.9999, 30 min = 1.599 ± 0.219 vs. vehicle; n.s; *p* > 0.9999, 1 h = 1.919 ± 0.537 vs. vehicle; n.s *p* > 0.9999, 2 h = 1.815 ± 0.530 vs. vehicle; n.s *p* > 0.9999) (Figure 6F). The lower p62 levels coincided with a trend for increased LC3-II levels, suggesting normal autophagy activation in the WT but not in the STING^−/−^ cells following H_2_O_2_ treatment. To confirm whether H_2_O_2_ treatment impairs autophagy activity in the STING^−/−^ cells, LAMP2 levels were measured. We detected sustained and increased LAMP2 expression in the STING^−/−^ cells challenged with 500 μM H_2_O_2_ across all time points (Figure 6G,H), with peak LAMP2 levels detected at 24 h (LAMP2 = 2.413 ± 0.489 vs. vehicle; * *p* < 0.05) while reduced in the WT MEF SV40 cells (LAMP2 = 0.808 ± 0.156 vs. vehicle, n.s *p* > 0.9999) (Figure 6I). The accumulation of LAMP2 expression suggests a block in the autophagosome–lysosomal degradation step, indicating impaired autophagy flux in the STING^−/−^ cells in response to H_2_O_2_.

The increased LC3-II/LC3-I levels detected at earlier time points in both WT and STING^−/−^ cells indicate a key role for STING in either inducing or blocking autophagic flux. To confirm whether STING is a key mediator of autophagy flux by H_2_O_2_, LC3-II levels in the presence or absence of a known autophagy inhibitor, bafilomycin A1 (BafA1), were analyzed (Figure 7A). If STING is an autophagy flux inducer, co-treatment with BafA1 and H_2_O_2_ would lead to higher LC3-II levels as compared to H_2_O_2_ alone. In contrast, if STING inhibited autophagy flux, LC3-II levels would remain unchanged [40,41]. WT cells treated for 3 h with H_2_O_2_ and BafA1 displayed higher LC3-II/LC3-I levels as compared to that of H_2_O_2_ alone (WT LC3-II/LC3-I ratio of H_2_O_2_ = 3.953 ± 0.781 vs. BafA1 ± H_2_O_2_ = 10.81 ± 2.061; ** *p* < 0.01) (Figure 7B), confirming that STING is an inducer rather than an inhibitor of autophagic flux. Similar treatment of BafA1 and H_2_O_2_ had no effect on LC3-II/LC3-I levels in the STING^−/−^ (STING**^−/−^** LC3-II/LC3-I ratio of H_2_O_2_ = 2.393 ± 0.621 vs. BafA1 ± H_2_O_2_ = 6.098 ± 1.052; n.s *p* > 0.9999) cells, confirming STING as the mediator of increased autophagy flux induced by H_2_O_2_. This result suggests that the increased LC3-II levels observed in H_2_O_2_-treated WT cells are due to increases in autophagosome formation rather than a block in autophagy flux. The increased LC3-II/LC3-I levels detected at earlier time points in both WT and STING^−/−^ cells would indicate a key role for STING in either inducing or blocking autophagic flux.

Interestingly, the addition of an autophagy inducer, rapamycin (Rap), and H_2_O_2_ in STING^−/−^ (STING**^−/−^** LC3-II/LC3-I ratio of VEH = 1.610 ± 0.256 vs. Rap ± H_2_O_2_ = 8.417 ± 1.479; ** *p* < 0.01) cells induced further accumulation of LC3-II levels which was similar to that observed in WT (WT LC3-II/LC3-I ratio of VEH = 3.138 ± 0.452 vs. Rap ± H_2_O_2_ = 11.63 ± 2.042; *** *p* < 0.001) cells, supporting a role for STING in mediating the autophagosome–lysosomal degradation step of autophagy. The reduced cellular viability observed in the H_2_O_2_-treated STING^−/−^ cells as compared to the WT cells can be attributed to the reduced protective effects of normal autophagy activity. Taken together, these results confirmed that autophagy is activated in response to H_2_O_2_ and identified STING as a critical mediator of efficient autophagy flux in this cellular stress context.

## 3. Discussion

The activation of the STING pathway and induction of type-I IFN signaling in response to H_2_O_2_-induced oxidative stress in vitro was confirmed in this study. Previously, the detrimental effects of the STING pathway in mediating the neuro-inflammatory response were demonstrated in a CCI-induced TBI model reported by our group [7]. Surprisingly, the results in this study suggest that the STING pathway can play a protective role, as demonstrated by a reduced cellular viability in the STING^−/−^ cells in response to H_2_O_2_ treatment. This finding proposes a previously unrecognized role for STING in modulating H_2_O_2_-induced oxidative stress and reveals a multifaceted role for STING in different disease/cellular models and stress-related pathophysiological processes.

### 3.1. STING Is Protective in H_2_O_2_ Induced Cell Death

Aberrant neuroinflammatory responses and increased oxidative stress are the major components that underlie many pathologies. However, the underlying molecular mechanisms contributing to oxidative stress involving STING are unknown. In this study, using a cell-based model of oxidative stress in vitro, a role for STING in response to H_2_O_2_-induced ROS generation was confirmed. STING is thought to regulate ROS production and oxidative stress through its anti-viral and anti-microbial activity [42]. It is known that an increased ROS level, generating oxidative stress and subsequent cell death, is a host defense mechanism to limit pathogen replication following infection [43,44]. Emerging evidence also supports a role for STING in inducing both apoptotic [45,46] and necroptotic cell death pathways [47,48]. However, evidence linking STING and the generation of ROS-induced oxidative stress is still lacking. More importantly, the underlying mechanisms that contribute to oxidative stress-induced cell death involving STING in a disease context, including central nervous system (CNS) injury, remains mostly unknown.

Intriguingly, the findings from this study identify a protective role for the STING pathway in H_2_O_2_-induced oxidative stress. It was found that 500 μM H_2_O_2_ induced higher production of intracellular ROS levels in the STING^−/−^ cells (Figure 5) which is associated with reduced pTBK1 (Figure 2A) and type-I (IFNα and IFNβ) levels (Figure 3C,D) as compared to WT cells. Importantly, the elevation in ROS levels corresponds with reduced cellular viability in the STING^−/−^ cells, suggesting STING (Figure 4A,B) may exert its protective effect by limiting the generation of detrimental ROS. Additionally, STING might also act as an antioxidant or play a role in regulating antioxidant components, such as superoxide dismutases (SODs), catalases, peroxidases and reductases, all of which eliminate excessive H_2_O_2_-induced ROS generation. However, the precise mechanisms by which STING protects against H_2_O_2_-induced oxidative stress in this study warrant further investigation.

### 3.2. The Type-I IFN and STING Pathways Are Activated after H_2_O_2_ Treatment

Furthermore, an increase in STING expression, both at the transcript and protein level, after H_2_O_2_ treatment was identified (Figure 1A,B). Increases in STING-mediated pTBK1 (Figure 2), IRF3 (Figure 3A), IFNα (Figure 3C) and IFNβ (Figure 3D) levels were also identified, confirming STING pathway activation after H_2_O_2_ treatment. These results, while not surprising, contradict the previous observations of STING in a TBI model. These discrepancies could be attributed to the complexity of the signaling pathways involved after TBI as compared to the simpler and controlled in vitro culture conditions that limit the activation of specific pathways induced by H_2_O_2_. Additionally, the peripheral-derived MEF cell line used in this study may elicit a different STING response as opposed to the complex interplay between neurons, microglia and astrocytes in the TBI brain. Henceforth, it can be postulated that the STING response may produce different outcomes in different cell types. Type-I IFNs can signal through the STING-TBK1-IRF3 axis induced by the nucleic acids, including cyclic–dinucleotide (CDN) or DNA, released by invading pathogens, resulting in the inhibition of viral and bacterial replication [49]. Recent studies have demonstrated STING-induced type-I IFN signaling through host-derived DAMPs, including mtDNA and self-DNA, in response to cellular injury [50,51]. Additionally, increased intracellular H_2_O_2_-induced ROS levels have been shown to promote the release of DAMPs, including ATP, uric acids, DNA, high mobility group box 1 (HMGB1) protein and mtDNA [52,53,54,55]. Henceforth, it is reasonable to postulate that the increases in STING (Figure 1) and IFNβ (Figure 3D) levels that coincide with reduced cellular viability (Figure 4B) can be attributed to DAMPs released from injured or dying cells after 500 μM H_2_O_2_ treatment. However, future studies confirming the exact molecules and underlying mechanisms leading to STING and type-I IFN signaling activation in this model are warranted.

It remains to be determined whether the downstream STING effectors, including increases in pTBK1, IRF3 and type-I IFN levels, contribute to the protective effects seen in the WT cells after 500 μM H_2_O_2_ treatment. However, activation of type-I IFN signaling mediated by the NF-κB transcription factor has been shown to promote cellular survival against pro-apoptotic stimuli [55,56] and STING is known to induce type-I IFN production through NF-κB activation [57]. Furthermore, it is interesting to note that increased STING and IFNβ mRNA levels showed a similar trend of activation (Figure 1A and Figure 3D) with the first wave of activation detected at 15 min and a second wave of activation detected at 24 h after 500 μM H_2_O_2_ treatment. Therefore, given that STING is protective after 500 μM H_2_O_2_ treatment, while activation of STING through DAMPs is consistent with a pro-death effect of STING reported in the literature [58,59], it is reasonable to postulate that STING may exert its protective effects through NF-κB-dependent type-I IFN signaling. However, future studies to confirm this are required.

### 3.3. Dual Role for STING in Regulating Autophagy Activity

The role of autophagy has been widely implicated under elevated oxidative stress with increased autophagic marker expression observed in disease pathogenesis [29,60]. However, its precise role and the mechanisms that trigger its induction and the involvement of the STING pathway remain unclear. Both the STING and type-I IFN pathways have emerged as critical players in autophagy activation in other cellular and disease models [61,62]. However, their interaction and regulation mechanisms following oxidative stress-induced cellular injury are unknown. Autophagy is a dynamic and complex cellular degradation process that requires careful analysis to identify and interpret its activity accurately. This study assessed hallmark markers of autophagy, including microtubule-associated protein 1 light chain 3 (LC3), SQSTM1/p62 and lysosomal-associated membrane protein 2 (LAMP2), in the H_2_O_2_-induced oxidative stress model. Conversion of LC3-I to LC3-II is representative of increased autophagy activation [63] while the degradation of p62 and LAMP2 at the late step of the autophagy process serves as a marker for normal autophagic flux. Given the observation of a protective effect in the WT cells following H_2_O_2_ treatment, this suggests that the increased LC3-II and p62 levels observed in STING^−/−^ cells do not indicate enhanced autophagy flux but rather impaired autophagy activity that contributes to cellular damage following elevated oxidative stress. Indeed, this study confirmed a protective role for STING following H_2_O_2_-induced oxidative stress in vitro. Evidence in the literature has also associated H_2_O_2_-induced oxidative stress with autophagy activation [58,64]. Here, we show H_2_O_2_-induced autophagy activation, with a trend for increased LC3-II and p62 levels in both H_2_O_2_-treated WT and STING^−/−^ cells. However, STING^−/−^ cells showed significantly increased and higher LAMP2 expression as compared to WT cells in response to H_2_O_2,_ suggesting a role for STING in regulating normal autophagy flux. It was also confirmed that increased and impaired autophagic activity, as measured by significant increases in LAMP2 protein levels, was evident in STING^−/−^ cells following H_2_O_2_ treatment. However, reduced LAMP2 levels in WT cells suggest an adaptation to normal autophagic activity in the presence of STING after H_2_O_2_ insult. This normal autophagic activity may serve as a protective mechanism to remove damaged cells and promote a protective environment, thus partially contributing to the reduced cell death observed in WT cells after H_2_O_2_ treatment.

The protective effect of STING in promoting cellular survival by maintaining normal autophagy activity after H_2_O_2_ treatment differs from its detrimental effects observed in our TBI model. However, it is not surprising as the increases in the normal autophagy response found in this study coincide with elevated intracellular ROS which may act as a defense mechanism to protect cells against ROS-induced oxidative stress, as reported in the literature [65,66]. It is also noteworthy that impaired autophagy activity in the STING^−/−^ cells correlated with reduced type-I IFN and STING-TBK1-IRF3 protein expression following H_2_O_2_ treatment. Given observations that the STING and type-I IFN pathways play a role in autophagy activation, these results highlight a critical role for STING in modulating normal autophagic activity rather than just as an autophagy inducer. Interestingly, a recent report demonstrated that cGAS-mediated autophagic activity protects cells from ischemia/reperfusion-induced oxidative stress independent of STING [67], while increased STING activation that is detrimental following TBI was found to be associated with its role in normal autophagy flux [7]. These discrepancies observed underscore our lack of understanding of how STING modulates autophagic activity in different disease models. Evidence in the literature has also associated H_2_O_2_-induced oxidative stress with autophagy activation [26,36]. It is possible, therefore, that STING may act as a molecular switch in promoting either detrimental or beneficial outcomes in TBI and H_2_O_2_-induced oxidative stress, respectively, by modulating autophagy activity. However, the underlying mechanisms leading to oxidative stress during CNS injury involving the STING and type-I IFN pathways remain to be investigated.

## 4. Materials and Methods

### 4.1. MTT Assay

The WT and STING^-/-^ MEF SV40 cells were a gift from Dr Kate McArthur (Walter and Eliza Hill Institute, Melbourne, Australia) and were cultured in Dulbecco’s minimal essential media (DMEM; Thermoscientific, Waltham, MA, USA) supplemented with 10% fetal bovine serum (FBS) and 5% penicillin/streptomycin (P/S; Thermoscientific, Waltham, MA, USA) at 37 °C/5% CO_2_. Cells were seeded at a density of 1 × 10^5^ cells/well in 6-well plates to assess cell viability by the conventional 3-(4,5-dimethylthiazol-2-yl)-2,5-diphenyltetrazolium bromide (MTT) reduction assay [68]. The WT and STING^-/-^ MEF SV40 cells were treated for the indicated times with H_2_O_2_ before being incubated with MTT (Sigma, St. Louis, MO, USA) solution (final concentration; 0.5 mg/mL) for 1 h at 37 °C/5% CO_2_. The medium was removed and 150 μL of dimethyl sulfoxide (DMSO) was added into each well, which was then read at 490 nm in a microplate reader. Results were expressed as percentages of the untreated control for that genotype.

### 4.2. DCF Assay

Intracellular ROS levels were quantitated using the 2′, 7′-dichlorofluorescin diacetate (DCFH-DA) assay (Sigma, St. Louis, MO, USA) [69]. WT and STING^-/-^ MEF SV40 cells were seeded at a density of 2 × 10^5^ cells/mL in a black 96-well plate with a transparent bottom and allowed to adhere overnight. Following H_2_O_2_ treatment (15 min to 48 h; 5 replicates for each genotype and time point), the cells were incubated with DCFH-DA (10 μM) for 30 min in the dark at 37 °C. Fluorescence was measured at excitation 485 nm/emission 535 nm using a microplate fluorescence reader (FlexStation^®^ 3 Benchtop Multi-Mode Microplate Reader, Molecular Devices, San Jose, CA, USA). The signal intensity was calculated relative to the genotype specific DMSO control group with the relative fluorescence intensity proportional to the intracellular ROS levels.

### 4.3. RNA Extractions and cDNA Synthesis

Cell lysates were homogenized in 1 mL Trizol (Invitrogen, Waltham, MA, USA) and incubated at room temperature for 10 min. 0.2 mL chloroform (Chem Supply, Melbourne, Australia) per 1 mL Trizol was added to the samples and they were centrifuged at 12,000× *g* for 15 min at 4 °C. The colorless, aqueous phase of each sample containing the RNA was transferred into a fresh 1.7 mL microcentrifuge tube. RNA was precipitated by adding 0.5 mL Propan-2-ol (Chem Supply) per 1 mL Trizol and the samples were centrifuged again at 12,000× *g* for 10 min at 4 °C. The supernatant from the tubes was discarded, and the RNA pellet was washed with 75% ethanol (Chem Supply, Melbourne, Australia) in diethyl pyrocarbonate (DEPC)-treated water (Sigma, St. Louis, MO, USA), vortexed and centrifuged at 7500× *g* for 5 min at 4 °C. The RNA pellet was air-dried and redissolved in RNAse-free H_2_O (Invitrogen, Waltham, MA, USA). The concentration and purity of the RNA samples was measured using the NanoDrop 1000 Spectrophotometer (Thermo Scientific, Waltham, MA, USA).

### 4.4. Quantitative Real Time Polymerase Chain Reaction (qPCR) Analysis

cDNA was transcribed from 1 μg RNA using a high capacity cDNA reverse transcription kit (Applied Biosystems, Waltham, MA, USA) as previously described [4]. Genes of interest were detected using Taqman (Applied Biosciences, Waltham, MA, USA) (Table 1) or SYBR green (GeneWorks, Thebarton, SA, Australia) (Table 2) primers. Ct values were obtained for each sample and relative transcript levels for each gene were calculated using the 2^−ΔΔCt^Method [70].

### 4.5. Western Blot Analysis

Protein concentrations were determined by Bradford assay (Bio-rad, Hercules, CA, USA) with 50 μg of protein lysed in 2× Novex^®^ Tris-glycine SDS sample buffer (Invitrogen, Waltham, MA, USA) for 10 min at 100 °C before resolution on 8% or 12% acrylamide SDS PAGE gels. Blots were then transferred to polyvinylidene fluoride (PVDF) membranes using a semi-dry transfer apparatus (BioRad, Hercules, CA, USA). Membranes were blocked in 5% skimmed milk in TBS-T for 1 h and incubated with primary antibodies (Table 3) in 2% skim milk in TBS-T at 4 °C overnight. Membranes were washed three times for 10 min each with TBS-T before being incubated with HRP-conjugated secondary antibodies (diluted in 2% skim milk in TBS-T) for 60 min at room temperature. Membranes were washed with TBS-T (3 × 10 min) and signals were detected using an ECL prime^®^ Western blotting detection kit (Amersham, Chicago, IL, USA) and visualized with the IQ350 imaging machine (GE Healthcare, Chicago, IL, USA). Post-image densitometry was performed using ImageJ software (NIH), whereby signal intensity was calculated in arbitrary units. For densitometry calculations, phosphorylation intensity was measured in arbitrary units and normalized to the β-actin loading control. These values were then calculated as fold change relative to untreated or vehicle control.

### 4.6. Statistical Analysis

Data are expressed as mean ± SEM and were analyzed using Graph Pad Prism 8.5 software. For qPCR and Western blot data, a one-way or two-way analysis of variance (ANOVA) was performed as appropriate, followed by a Bonferroni’s post-hoc analysis, with a value of *p* < 0.05 considered statistically significant.

## 5. Conclusions

This study provides evidence to suggest a novel role for STING in mediating the type-I IFN pathway in a model of elevated oxidative stress. This finding proposes a previously unrecognized role for STING in modulating H_2_O_2_-induced oxidative stress and reveals the multifaceted role of STING in different disease/cellular models and elevated oxidative stress-related pathophysiological processes.

## Figures and Tables

**Figure 1 ijms-21-07059-f001:**
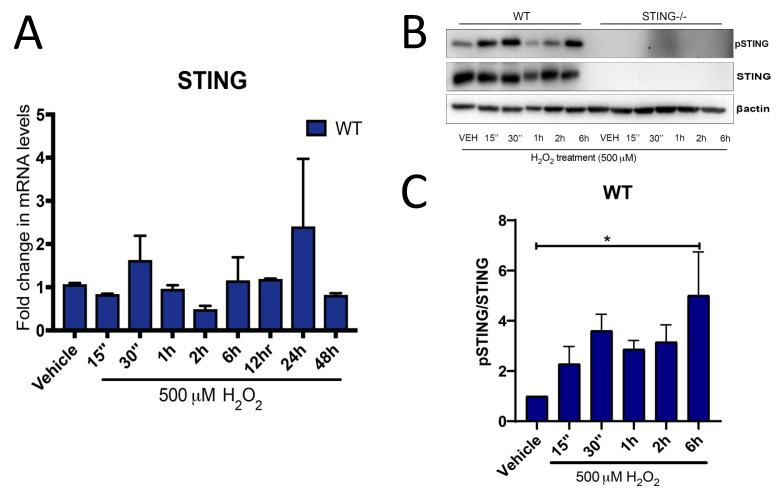
H_2_O_2_ induces an upregulation in stimulator of interferon genes (STING) expression. Increased STING expression was detected by qPCR analysis (**A**) (*n* = 4 for each time point) in WT MEF SV40 cells exposed to H_2_O_2_ treatment as compared to vehicle control. Increased phosphorylation of STING (pSTING) was confirmed at the protein level by western blot analysis (**B**) with a lack of STING expression identified in STING^−/−^ cells (representative blot of *n* = 5 blots). Quantification of pSTING expression relative to total STING in (**B**) is shown in (**C**). All data are expressed as mean ± SEM, * *p* < 0.05.

**Figure 2 ijms-21-07059-f002:**
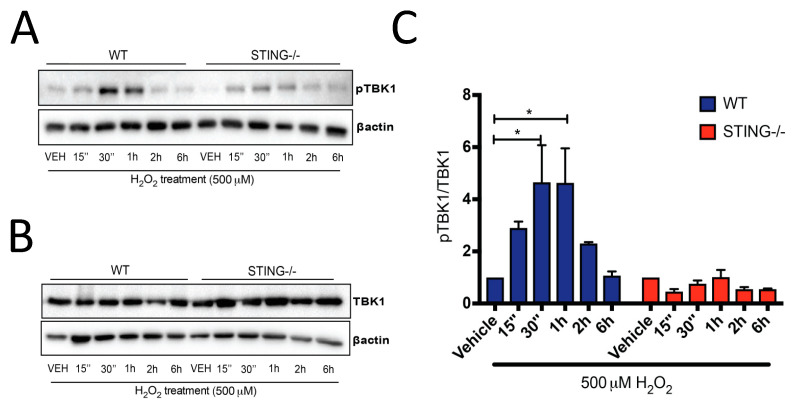
Activation of the downstream mediator of STING, TBK1, is increased following H_2_O_2_ treatment. Western blot analysis confirmed increased phosphorylation of TBK1 (pTBK1) (**A**) in WT MEF SV40 cells following treatment with 500 μΜ H_2_O_2,_ with levels unchanged in STING^−/−^ MEF SV40 cells (representative blot of *n* = 4 blots). Total TBK1 expression (**B**) was unchanged by H_2_O_2_ treatment in both cells (representative blot of *n* = 4 blots). Quantification of (**A**) and (**B**) is shown in (**C**), demonstrating an increase in pTKB1 expression relative to total TBK1 levels in H_2_O_2_-treated WT MEF SV40 cells. Data represent the mean ± SEM, * *p* < 0.05.

**Figure 3 ijms-21-07059-f003:**
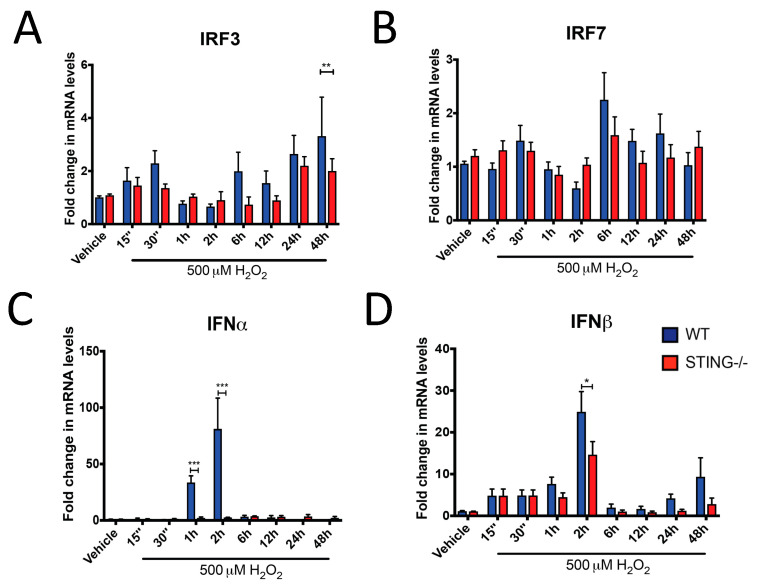
Type-I interferon (IFN) signaling is upregulated by H_2_O_2_ in a STING-dependent manner. WT and STING^−/−^ cells were treated with 500 μM H_2_O_2_ for the indicated time points before qPCR analysis was performed to determine (**A**) IRF3, (**B**) IRF7, (**C**) IFNα and (**D**) IFNβ mRNA levels (*n* = 4). Data represent mean ± SEM, * *p* < 0.05, ** *p* < 0.01, *** *p* < 0.001.

**Figure 4 ijms-21-07059-f004:**
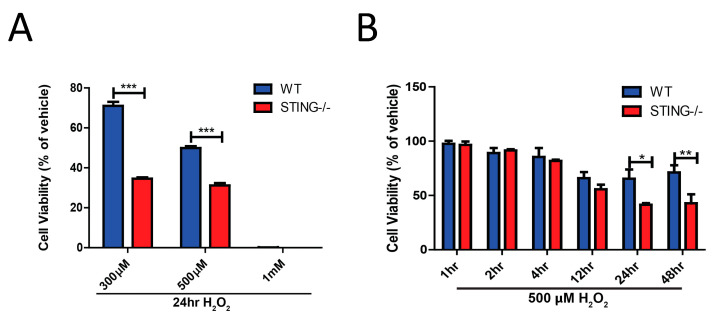
Genetic deletion of STING is detrimental to cellular survival following H_2_O_2_ treatment. MTT assay identified reduced cellular viability in STING^−/−^ cells as compared to WT cells following (**A**) dose-dependent H_2_O_2_ concentrations for 24 h, (**B**) a time-course of 500μM H_2_O_2_ concentration (*n* = 4). Data represent mean ± SEM, * *p* < 0.05, ** *p* < 0.01, *** *p* < 0.001.

**Figure 5 ijms-21-07059-f005:**
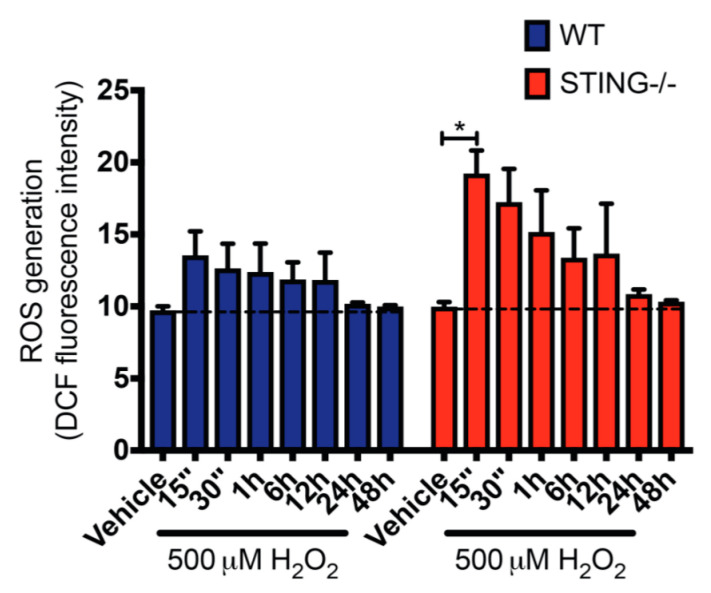
STING^−/−^ cells display greater ROS production in response to H_2_O_2_ treatment compared with WT cells. WT and STING^−/−^ cells were treated with 500 μM H_2_O_2_ for the indicated time points before intracellular ROS levels were determine by DCF assay. Data represent mean ± SEM, * *p* < 0.05, *n* = 5.

**Figure 6 ijms-21-07059-f006:**
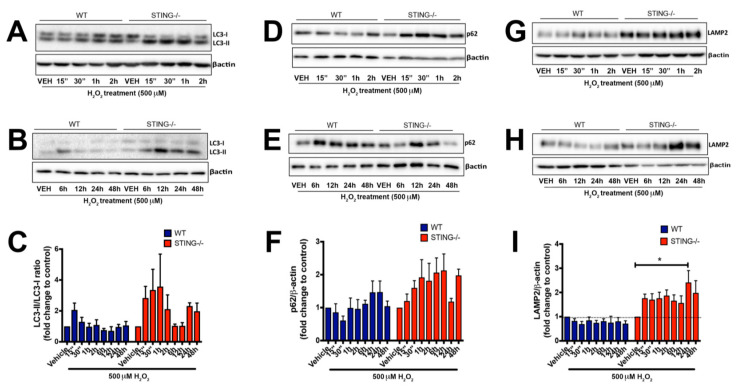
STING^−/−^ cells display increased accumulation of autophagy markers following H_2_O_2_ treatment. WT and STING^−/−^ cells were treated with 500 μM H_2_O_2_ for 15 min to 2 h (**A**,**D**,**G**) and 6 h to 24 h (**B**,**E**,**H**), and LC3, p62 and LAMP2 protein expression was analyzed by Western blot analysis (representative blot of *n* = 4 blots). LC3-I, LC3-II, p62 and LAMP2 levels were normalized to β-actin levels (**C**,**F**,**I**). For densitometry calculations, the LC3-II/LC3-I ratio was then determined from these values and was calculated as a fold change relative to the vehicle control as shown in (**C**). Data are expressed as mean ± SEM. (“) = min. * *p* < 0.05.

**Figure 7 ijms-21-07059-f007:**
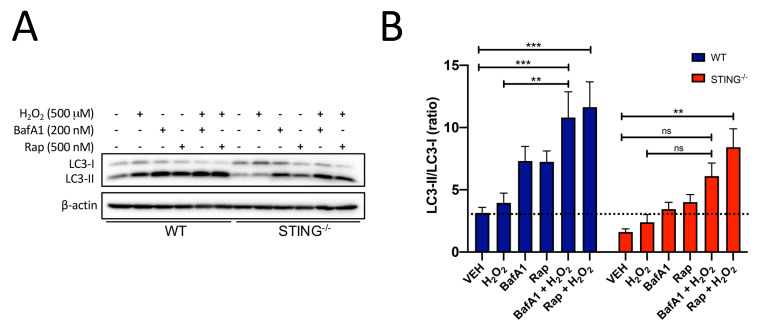
Presence of STING increases autophagy flux in response to H_2_O_2_ treatment. WT and STING^−/−^ cells were treated with 500 μM H_2_O_2_ alone or in the presence of 200 nM bafilomycin or 500 nM rapamycin for 3 h. (**A**) Western blot analysis was performed with an LC3 antibody with β-actin as a loading control (representative blot of *n* = 6 blots). (**B**) LC3-II and LC3-I levels were normalized to β-actin levels; an LC3-II/LC3-I ratio was then determined from these values. Data are expressed as mean ± SEM, ** *p* < 0.01, *** *p* < 0.001. n.s *p* > 0.05.

**Table 1 ijms-21-07059-t001:** Taqman primers used for qPCR analysis.

Gene	Species	Refseq	Amplicon Length (bp)	Catalogue No
GAPDH	Mouse	NM_008084.2	107	Mm99999915_m1
IFNβ	Mouse	NM_010510.1	69	Mm00439552_s1
IRF3	Mouse	NM_016849.4	59	Mm00516779_m1
IRF7	Mouse	NM_001252600.1	67	Mm00516788_m1
		NM_001252601.1		
		NM_016850.3		
STING	Mouse	NM_028261.1	173	Mm01158117_m1

**Table 2 ijms-21-07059-t002:** Sybr green primers used for qPCR analysis.

Gene	Forward Primer (5′-3′)	Reverse Primer (5′-3′)
GAPDH	ATCTTCTTGTGCAGTGCCAGC	ACTCCACGACATACTCAGCACC
IFNα	GCAATCCTCCTAGACTCACTTCTGCA	TATAGTTCCTCACAGCCAGCAG
IFNαE4	-	TATTTCTTCATAGCCAGCTG

**Table 3 ijms-21-07059-t003:** Primary antibodies used for Western blot analysis.

Primary Antibodies	Origin	Dilution	Company	Catalogue No
anti-pSTING (s365)	rabbit	1 in 500	Cell Signaling (Danvers, MA, USA)	72971
anti-STING	rabbit	1 in 1000	Cell Signaling (Danvers, MA, USA)	13647
anti-LC3	rabbit	1 in 1000	MBL (Woburn, MA, USA)	PM036
anti-SQSTM1/p62	mouse	1 in 1000	Abcam (Cambridge, UK)	ab56416
anti-β-Actin	mouse	1 in 1000	Sigma-Aldrich (St. Louis, MO, USA)	A5441
anti-LAMP2	rat	1 in 1000	Abcam (Cambridge, UK)	ab25339
anti-pTBK1(s172)	rabbit	1 in 1000	Abcam (Cambridge, UK)	ab109272
anti-NAK/TBK1	rabbit	1 in 1000	Abcam(Cambridge, UK)	ab40676

Secondary antibodies used; horseradish peroxidase conjugated goat anti-rabbit (1: 1000, Dako, P0488), goat anti-mouse (1:1000, Dako, P0447) and rabbit anti-rat (1:1000, Abcam, ab6734).

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
