# Peer review of "STING-Mediated Autophagy Is Protective against H2O2-Induced Cell Death"

_ijms, 2020, doi:10.3390/ijms21197059_

Round 1

Reviewer 1 Report

DearAuthor

The manuscript titled STING-mediated autophagic flux is protective against H2O2-induced cells death was studied by Amar Abdullah et al is well designed and informative.

There are some minor changes before publication.

  1. I am not fine with the titled STING mediated autophagic flux is protective against H2O2-induced cell death. The reason, autophagy is a complex process and the term autophagy flux is a complete process initiation, elongation, and degradation. You have shown LC3B, P62 and LAMP2 expression and this is not sufficient to confirm the autophagy flux. In general, p62 will be degraded in the lysosomal degradation and here the author showed increased p62 levels in STING negative cells.
  2. How did you arrive IFN signaling is upregulated by H2O2.
  3. Do you have any supporting experimental design for ROS generation?
  4. Fig 6 (A-I), labeling is not clear.
  5. There are some typo errors that will be considered before publication. for eg. H2O2 is written H202. 

Author Response

The manuscript titled STING-mediated autophagic flux is protective against H2O2-induced cells death was studied by Amar Abdullah et al is well designed and informative.

There are some minor changes before publication.

  • I am not fine with the titled STING mediated autophagic flux is protective against H2O2-induced cell death. The reason, autophagy is a complex process and the term autophagy flux is a complete process initiation, elongation, and degradation. You have shown LC3B, P62 and LAMP2 expression and this is not sufficient to confirm the autophagy flux. In general, p62 will be degraded in the lysosomal degradation and here the author showed increased p62 levels in STING negative cells.

Reply: The title of the manuscript has been updated to "STING mediated autophagy is protective against H2O2-induced cell death. 

  • How did you arrive IFN signaling is upregulated by H2O2.

Reply: IFN signaling was shown to be upregulated by H2O2 in Figure 3 with an upregulation in mRNA levels of the type-I IFNs, IFNalpha and IFNbeta, and the modulator, IRF3.  Expression was reduced in STING-/- cells suggesting this H2O2-induced increase in IFN signaling was through a STING-IRF3 mediated pathway.

  • Do you have any supporting experimental design for ROS generation?

Reply: The experimental design for ROS generation was based on a previous publication from our lab (Loh et al., 2009; Cell Metabolism).  This reference has now been included in the methods section.

  • Fig 6 (A-I), labeling is not clear.

Reply: The labelling on Fig 6 (A-I) has been enlarged and is now clearer.

  • There are some typo errors that will be considered before publication. for eg. H2O2 is written H202. 

Reply: The manuscript has undergone additional editing and any typos corrected.

Reviewer 2 Report

The authors devoted their work to studying new mechanisms of STING1 involvement in the cell response to oxidative stress induced by H2O2. These studies have demonstrated very important data on the involvement of STING in the induction of autophagy. The presented results are of both fundamental and applied importance. I believe that the work done by the distinguished authors can be published.

Author Response

No response to the reviewer's comments were required with no improvements or alterations suggested.